# Eight years of community structure monitoring through recreational citizen science at the "SS Thistlegorm" wreck (Red Sea)

Chloe Lee[1], Erik Caroselli[1,2]*, Mariana Machado Toffolo[1,2], Arianna Mancuso[1,2], Chiara Marchini[1,2], Marta Meschini[1,2], Stefano Goffredo[1,2]

**1** Department of Biological, Marine Science Group, Geological and Environmental Sciences, University of Bologna, Bologna, Italy, **2** Fano Marine Center, The Inter-Institute Center for Research on Marine Biodiversity, Resources and Biotechnologies, Fano, Italy

* erik.caroselli@unibo.it

**Data Availability Statement:** Relevant data are publicly available in the figshare database (doi.org/10.6084/m9.figshare.21197269).

## Abstract

Large artificial coral reef communities, such as those thriving on sunken shipwrecks, tend to mirror those of nearby natural coral reefs and their long-term dynamics may help future reef resilience to environmental change. We examined the community structure of the world-renown "SS Thistlegorm" wreck in the northern Red Sea from 2007 through 2014, analyzing data collected during the recreational citizen science Red Sea monitoring project "Scuba Tourism for the Environment". Volunteer divers collected data on 6 different diving parameters which included the date of the dive, maximum depth, average depth, temperature, dive time, hour of dive, and gave an abundance estimation of sighted taxa from a list of 72 target taxa. Although yearly variations in community structure were significant, there was no clear temporal trend, and 71 of all 72 target taxa were sighted throughout the 8 years. The 5 main taxa driving variations among year clusters in taxa presence/absence (Soft Tree Coral—*Dendronephthya spp.*, Giant Moray—*Gymnothorax javanicus*, Squirrel Fish—*Sargocentron spp.*, Humpback Batfish—*Platax spp.*, and Caranxes—Carangidae) and taxa abundance (Soft Tree Coral, Giant Moray, Red Sea Clownfish—*Amphiprion bicinctus*, Napoleon Wrasse—*Cheilinus undulatus*, and Caranxes) data were determined. The "SS Thistlegorm" provides a compelling example of how artificial coral reefs can sustain a well-established community structure similar to those of their natural counterparts.

## Introduction

The biological communities of tropical and sub-tropical coral reefs support some of the highest biodiversity in the world [1] and provide a wide array of socio-economic services including coastal protection, water quality and chemical cycling, fisheries and materials markets (e.g. sponges, ornamental use, medicine, etc), and experiential benefits (e.g. education, recreation) [2]. Despite their assortment of values, coral species homogenization [3], species extinctions

**Funding:** STE project was funded by Project AWARE Foundation, ASTOI Association, Ministry of Tourism of the Arab Republic of Egypt, Settemari S.p.A Tour Operator, Scuba Nitrox Safety International, Viaggio nel Blu Diving Center. The funders had no role in study design, data collection and analysis, decision to publish, or preparation of the manuscript.

**Competing interests:** The authors have declared that no competing interests exist.

[4], and overall changes of coral communities are expected in future [5]. In fact, the Intergovernmental Panel on Climate Change projections [6] and current research indicate a multitude of threats to coral reefs including but not limited to ocean warming and acidification, overfishing, and rise in coastal human populations and impacts [7].

One of the largest coral reef systems of the world, with notably high endemism, lies within the Red Sea [8]. Due to its peculiar geo-evolutionary past resulting in large latitudinal gradients of sea surface temperature and salinity [8] the Red Sea provides an attractive location for studying the response of coral reefs to threats of their projected immediate future. The northern Red Sea is considered a coral refuge because, although it has experienced some of the highest sea surface temperature anomalies, there have been very few coral bleaching events compared to its southern portion [7, 9]. Notwithstanding the relative resistance of reefs in the northern Red Sea, there is still evidence of general coral colony size decline and species homogenization throughout [10], having the potential to induce changes in the biological communities they support. The importance of monitoring community variation lies not only in its fundamental value but also the economic income from fisheries [11] and especially the tourist industry, which provide an array of socio-economic opportunities for many coastal countries [12]. Unfortunately, there is general lack of information regarding Red Sea coral communities, which are relatively understudied when compared to the Great Barrier Reef or reef systems of the Caribbean Sea [13]. This can pose conservation challenges when attempting to monitor and analyze crucial changes in the biological communities of the Red Sea.

An innovative and upcoming approach to gathering large amounts of data in a time efficient way is the practice of citizen science. This approach is not so novel in terms of human history, but current technological advances of citizen science in the spreading and sharing of information and data [14, 15] have benefits and uses that are now an integral part of modern science [16]. Citizen science can be a critical asset in helping researchers, policy makers, and stakeholders overcome resource limitations (e.g., economical, temporal, geographical). Community-based management, and community-based monitoring are two of the main methodologies of the citizen science approach. The former involves direct participation of citizens and stakeholders in management decisions [17], while the latter implies the collaboration of citizens and stakeholders for data collection to monitor, track, and respond to areas of concern (e.g., environmental health) [18]. Scuba Tourism for the Environment, which was launched in 2007 (STE; www.steproject.org), provides a great example of involving large numbers of volunteers for community-based monitoring throughout much of the Red Sea [19]. Divers participating in the STE program collected ecological community data with the assistance of active dive centers and trained guides throughout Egypt. Recreational citizen science was applied throughout the project. This approach to citizen science allows participants to carry out their normal activities (volunteer behavior is unchanged throughout the survey), and to collect casually observed data. To analyze the reliability of data collected by the participants, data were compared with those collected by control divers (marine biologists of the Marine Science group at the University of Bologna) [20]. Consistency, or the similarity of data collected by individual volunteers during the same dive, was the lowest ranking parameter. The percentage recorded of the total number of taxa present (acquired from control diver data), or the Percent Identified, was the highest-ranking parameter [20]. This indicates that while divers can accurately identify most taxa there is a tendency to focus on certain organisms due to personal interest. However, with correct development to the needs of specific projects, data collected through citizen science can reliably support monitoring efforts [16, 17, 20]. The SS Thistlegorm, a world class dive site [21] and one of topmost visited wreck dives in the world [22], was one of the dive locations monitored in the STE project and is the focus of this study. The SS Thistlegorm was built in 1940 as a steal cargo steamship and was quickly taken over for the

war efforts of the British military. On her final voyage from Glasgow (UK) to Alexandria (Egypt) she was met with two German Heinkel HE 111 bombers. The 128 m long and 17 m high cargo steam ship was split in two by munitions explosions and sunk within minutes [23]. Her ultimate resting ground is in the Straights of Gubal 30 meters below the sea surface [23]. In the 80 plus years since the discovery of the wreck by Jacques-Yves Cousteau in 1955, and the development of the Sharm el-Sheikh resort in the 1990s, the SS Thistlegorm has recruited up to 175,000 visiting divers per year from around the world [23]. Due to its high popularity and frequent visitors, there are rising concerns regarding the structural longevity of this historic site. Irresponsibly moored boats, inexperienced divers, and even air bubbles can cause irreparable damage to the structural integrity of underwater relics [24] as well as the biological community residing there [25]. Additionally, because the wreck is afforded no legal protection, it is prey to looting and souvenir plundering (www.thethistlegormproject.com). A dedicated team of divers, archeologists, and researchers developed the Thistlegorm Project (www.thethistlegormproject.com) to survey the site using 360 videography for the purposes of creating an accurate archeological 3D survey and to raise awareness for the protection of underwater cultural heritage sites.

Aside from the historical importance of the SS Thistlegorm wreck, the ship has recruited numerous marine life forms through the decades spent on the sandy bottom close to a natural coral reef and far from touristic resorts (70 km by boat from Sharm el Sheikh). We report here a first investigation on the temporal community trends of biological organisms at the SS Thistlegorm wreck. Scientific literature concerning long-term (more than 5 years) temporal trends and monitoring of community structure on wreck and/or artificial reefs is scarce [26]. There is extensive interest, however, in the role of artificial reefs, (including wrecks and scuttled ships) as a conservation tool, potentially serving as a compensatory habitat for anthropogenically damaged natural reefs [27–29], and as possible refugia assisting corals in the colonization of cooler waters in response to ocean warming [26]. Shipwrecks in particular have been shown to significantly increase biodiversity of both fish and benthic species on soft bottom environments [30, 31] along the coast and represent key microhabitats in the open ocean [32]. The material of the wreck, the bottom type it lies upon, as well as its structural complexity play key roles in determining marine community and have been shown to be fundamental in epibenthic colonization patterns [33].

The aim of this study was to characterize the community structure and its temporal trends of the SS Thistlegorm, over 8 years of monitoring through recreational citizen science.

## Materials and methods

### The study site

The SS Thistlegorm lies 27˚ 48' 30.59" N and 33˚ 55' 7.19" E at 30 m depth on sandy bottom substrate within the Straights of Gubal. The bow of the ship faces S40˚W, with tidal currents flowing from Northeast to Southwest at varying strength, from stern to bow [34]. Her final voyage entailed a resupply mission to the British Armed Forces and was destined for Alexandria (Egypt). The cargo onboard the SS Thistlegorm included Bedford trucks, BSA motorcycles, Wellington rubber boots, aircraft parts, rifles, land mines, ammunition, and other weaponry, as well as two LMS Stanier Class 8F steam locomotives among other goods [28], much of which is potentially hazardous to the marine environment.

### Data collection and isolation

The Scuba Tourism for the Environment—Red Sea Biodiversity Monitoring Program (www.steproject.org),included the collection of large scale spatial-temporal biodiversity data within

the Red Sea from 2007 to 2015 using a recreational citizen science approach where the behavior of the underwater tourists is unaltered during surveys [19]. Participant volunteers completed questionnaires that contained general demographic information (name, country of residence, address, diving certification, etc.), dive site and environment type (sandy bottom, wreck, blue, or coral reef), six diving parameters (date (PY), maximum depth (MD), depth where they spent most of their time (AD), temperature (T), dive time (RTD), and hour (FD)), and the sightings of 72 target taxa which were recorded on the questionnaires immediately following the dive and without the use of underwater marking materials during the dive [19]. For each of the taxa sighted, the divers also recorded an estimated Sighting Abundance according to 3 classes: 1, 2, and 3 (rare, frequent, and very frequent respectively) [19]. The classes were weighted to each taxon's individual expected occurrence [19]. The 72 faunal taxa were chosen because they are representative of the main ecosystem trophic levels within the Red Sea, they are common/abundant, and they are easily identifiable by recreational divers [19]. Data coming from the Thistlegorm wreck site were aggregated per year.

## Preliminary treatment and analyses

Seven of the dives within the Thistlegorm dataset were viable for validation trials and were analyzed to measure the quality of volunteer collected data. The mean similarity index, accuracy, and correctness of abundance ratings (CAR) parameters were tested in accordance with the standard methodology for this citizen science project [20].

Thistlegorm data were then split into three data sets: 1) taxa abundance; 2) taxa presence/absence (p/a), obtained through an overall transformation of taxa abundance; and 3) diving parameters. The sighting frequency (*SF%*) and Relative Abundance (*RA*) for each taxon were calculated as such [19]:

$$SF\% = \left( \frac{number\ of\ sightings\ per\ taxon}{number\ of\ dives} \right) x\ 100$$

$$RA = (SF\%)(Average\ Sighting\ Abundance)$$

The analyses below were completed using PRIMER-E version 6 with PERMANOVA+ version 1 software (PRIMER-E, Ltd., Ivybridge, UK).

For a general picture of the trophic community composition and to identify outliers within each year, 2-dimensional plots were created using non-metric multidimensional scaling (MDS) from the Bray-Curtis resemblance matrices of both taxa data sets [35, 36]. A BVSTEP "Best" test was performed on both taxa data sets to find the subset of taxa that best represented the community of each year [37]. These subsets of taxa were later used for aggregation of the data in the temporal analysis.

A distance matrix of normalized diving parameters was created using Euclidean distances. Both taxa distance matrices were tested for a relationship with the distance matrices of the diving parameters using the Relate test to determine whether the variation between years was affected only by the change in time or also by diving parameters [38]. Additional distance based linear models [39] were ran for each year with a significant relationship between the taxa data and diving parameters, providing the individual parameters that significantly affected the variation of community structure in each year.

To test the differences in diving parameters over the years, a permutational multivariate analysis of variance (PERMANOVA, [39]) was designed with the fixed factor of "year" and run on the Euclidean resemblance matrix of the normalized aggregate diving parameters. Both main and pairwise tests were run. Additional PERMANOVAs were created for individual

diving parameters to test whether each different diving parameter was significantly different among the years.

## Temporal analysis

Two reduced datasets containing only the BVSTEP best subsets of taxa representing th community structure from any individual year were obtained, and resemblance matrices were created from Bray-Curtis distances of both the abundance taxa data and the overall-transformed p/a taxa data [35]. To test significant difference among years, PERMANOVAs [39] were run with the fixed factor "year". For a sharper visualization of temporal variation in community structure, distances between centroids were obtained from the resemblance matrices using "Year" as the grouping factor and visualized with 2D and 3D MDS plots. 3D plots were used to better interpret the direction of vectors in 2D plots and were not included as figures in the current manuscript. Any similar groupings of the centroidal "years" were found with a hierarchical cluster analysis [40]. The average sample taxa abundance and presence/absence from each year were calculated, and a BVSTEP Best analysis was run from the Bray-Curtis resemblance matrix to find the subset of taxa that best explained the trohic community structure variance within centroid data sets [37].

## Ethics statement

Participants (or parents/guardians in case of minors) gave their informed, written consent by signing a declaration inserted in the questionnaires. STE project and its consent acquisition procedure have received the approval of Bioethics Committee of the University of Bologna (prot. 2.6).

## Results

### Data collection and isolation

Nine questionnaires from 2015 were removed from the analysis, as per lack of efficacy for less than 10 questionnaires per year (following 19). While Branchini and colleagues [19] examined the "coral reef" environment questionnaires, this study reports findings from an unpublished data set within the "wreck" environment type. Data from a total of 390 questionnaires between 2007 and 2014 were included in this analysis (Table 1). Questionnaire counts per year ranged from 12 to 84, and significantly decreased over time ($R^2 = 0.78$, $P < 0.05$). Over the 8 years, 71 (97%) of the 72 target taxa present on the STE questionnaire were sighted (all taxa excluding the Manta Ray).

### Preliminary treatment and analysis

According to the RELATE test, years 2007 ($\rho_s = 0.162$, N = 6, $P = 0.007$), 2008 ($\rho_s = 0.157$, N = 2, $P = 0.003$), 2009 ($\rho_s = 0.111$, N = 45, $P = 0.046$), and 2012 ($\rho_s = 0.217$, $N = 9$, $P = 0.01$) showed a significant relationship between taxa abundance and dive parameters (Table 2; S1 Table). In all other years there was no significant relationship (Table 2; S1 Table). The DistLM of taxa abundance data in 2007 showed a significant relationship ($P < 0.05$) with all 6 diving parameters: date (PY), maximum depth (MD), depth where they spent most of their time (AD),

**Table 1. Yearly counts of completed questionnaires.**

| Year | 2007 | 2008 | 2009 | 2010 | 2011 | 2012 | 2013 | 2014 | Total |
|---|---|---|---|---|---|---|---|---|---|
| Number of Questionnaires | 84 | 62 | 59 | 59 | 72 | 29 | 12 | 13 | 390 |

**Table 2. SS Thistlegorm dive parameters with significant ($P < 0.05$) sequential DistLM test to community structure by year for abundance and presence/absence data.** PY as date (expressed as the percentage of year), MD as maximum depth, AD as depth at most time spent, T as temperature, RTD as dive duration, and FD as hour (expressed as percentage of day).

| Year | Abundance | Presence/Absence |
|---|---|---|
| 2007 | PY, MD, AD, T, RTD, FD | PY, AD, T, RTD, FD |
| 2008 | PY, MD, AD, T, RTD | PY, MD, T, RTD |
| 2009 | PY, AD, FD | |
| 2010 | | |
| 2011 | | |
| 2012 | PY, T | PY, T, RTD |
| 2013 | | |
| 2014 | | |

temperature (T), dive duration (RTD), and hour (FD) (Table 2; S2 Table). In 2008 all the diving parameters except FD were significantly correlated ($P < 0.05$) to the taxa abundance (Table 2; S2 Table). PY, AD, and FD were significantly related ($P < 0.05$) to taxa abundance in 2009 (Table 2; S2 Table). In 2012 only PY and T diving parameters were significantly related to the community structure ($P < 0.05$; Table 2; S2 Table).

Regarding taxa presence/absence (p/a) data, the RELATE results indicated that in 2007 ($\rho_s$ = 0.18, N = 1, $P$ = 0.002), 2008 ($\rho_s$ = 0.136, N = 13, $P$ = 0.014), and 2012 ($\rho_s$ = 0.209, N = 4, $P$ = 0.005) there was a significant relationship between the community structure and dive parameters (Table 2, S1 Table). The DistLM in 2007 revealed that all dive parameters except MD had significant relationships to the taxa p/a data ($P < 0.05$; Table 2, S3 Table). In 2008 PY, MD, T, and RTD were significant ($P < 0.05$; Table 2, S3 Table), while only PY, T and RTD resulted significant in 2012 ($P < 0.05$).

Aggregated diving parameters showed significant differences among years (PERMANOVA, F = 9.25, df = 7, $P$ = 0.001). PY, MD, AD, T and RTD all differed significantly among years ($P < 0.05$). FD was homogenous among years.

Of the 7 validation trial dives, the average mean similarity index score was 98.8%, the average mean accuracy score was 57.1% and ranged from 40.4 to 70.0%, and the mean average CAR was 73.8%.

## Temporal analysis

According to the PERMANOVA from both the taxa abundance and p/a data, the community structure was significantly different between each pair of years (F = 4.88, df = 7, $P$ = 0.001 and F = 4.88, df = 7, $P$ = 0.001 respectively). The variation in community structure of taxon abundance and p/a through years 2007–2014 was represented as an MDS (Figs 1A and 2A). Given the high stress value ($> 0.3$) of these MDS plots, further centroidal MDS plots of both taxa data sets were generated to have a clearer, even if more general, visualization highlighting the similar groupings of years within their respective distances resulting from the hierarchical cluster analysis (Figs 1B and 2B).

The BVSTEP subset of taxa that explained the distances between yearly centroids of community structure for the taxa abundance data were the Soft Tree Coral (*Dendronephthya spp.*,), Giant Moray (*Gymnothorax javanicus*), Red Sea Clownfish (*Amphiprion bicinctus*), Napoleon Wrasse (*Cheilinus undulates*), and Caranxes (Carangidae). The BVSTEP best subset of taxa explaining the distances between yearly centroids of community structure for the taxa p/a data were the Soft Tree Coral, Giant Moray, Squirrel Fish (*Sargocentron spp.*), Humpback Batfish

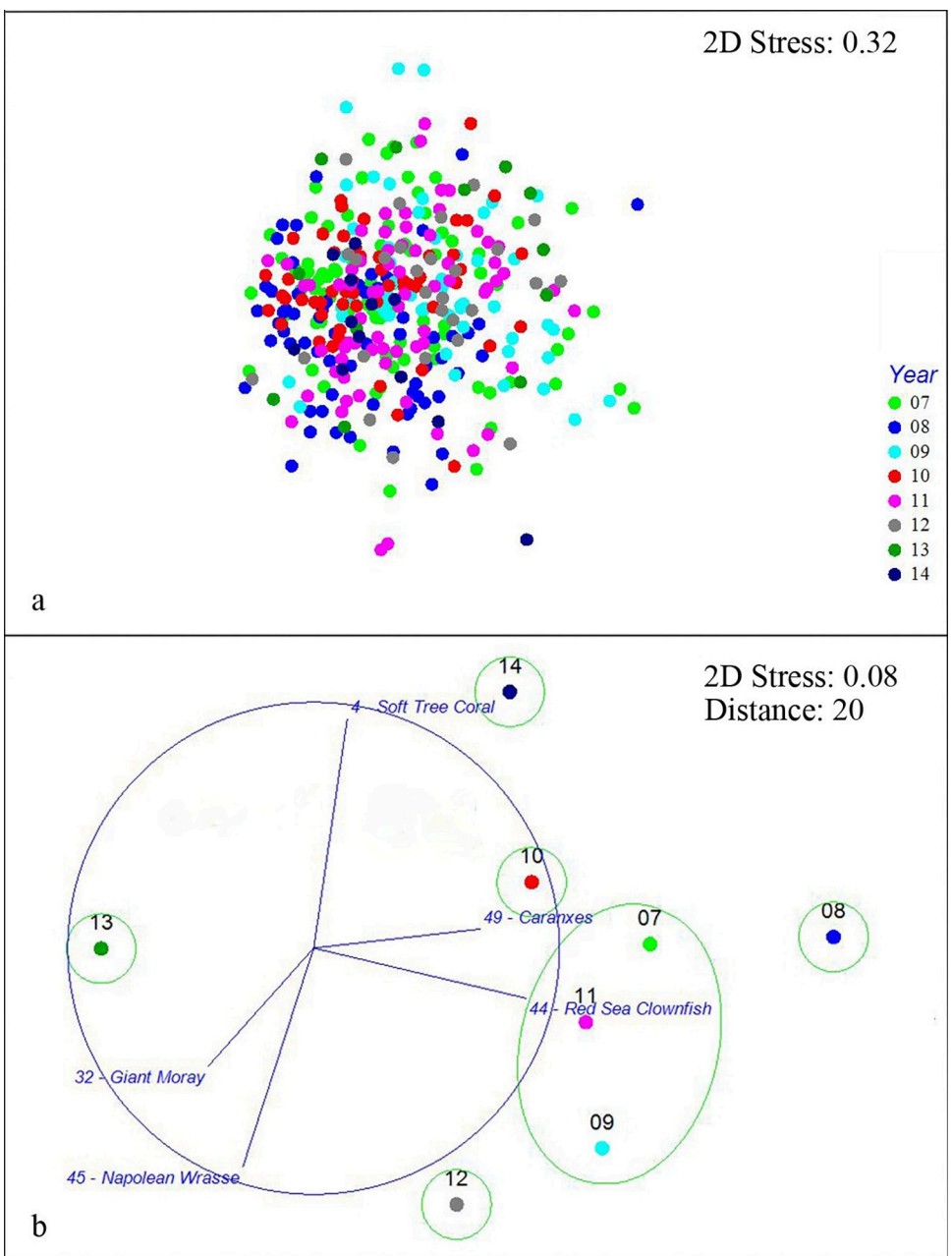

**Fig 1. a) MDS plot of the SS Thistlegorm taxa abundance community structure over 8 years.** Each point represents an individual questionnaire while the different shapes/colors indicate the different years. **b) Centroidal MDS of SS Thistlegorm taxa abundance community structure variation over 8 years with overlayed vectors of aggregated best taxa and the indicated trajectories from 2007–2014.** Each point represents the central location of the groups "Year". The green circles surround the groups resulting from the hierarchical cluster analysis.

(*Platax batavianus*), and Caranxes. Tables 3 and 4 show the relative abundance and SF% of the best taxa through the 8 years.

Years 2007, 2009, and 2011 cluster around a general community structure in the taxa abundance data (Fig 1B). The Red Sea Clownfish (*Amphiprion bicinctus*) was overrepresented in 2008 (98 in 2008 *vs* 49–80 in 2007, 2009, and 2011; Table 3). The year 2010 was overrepresented in Soft Tree Coral (*Dendronephthya spp.*,) (98 in 2010 *vs* 41–77 in 2007, 2009, and 2011;

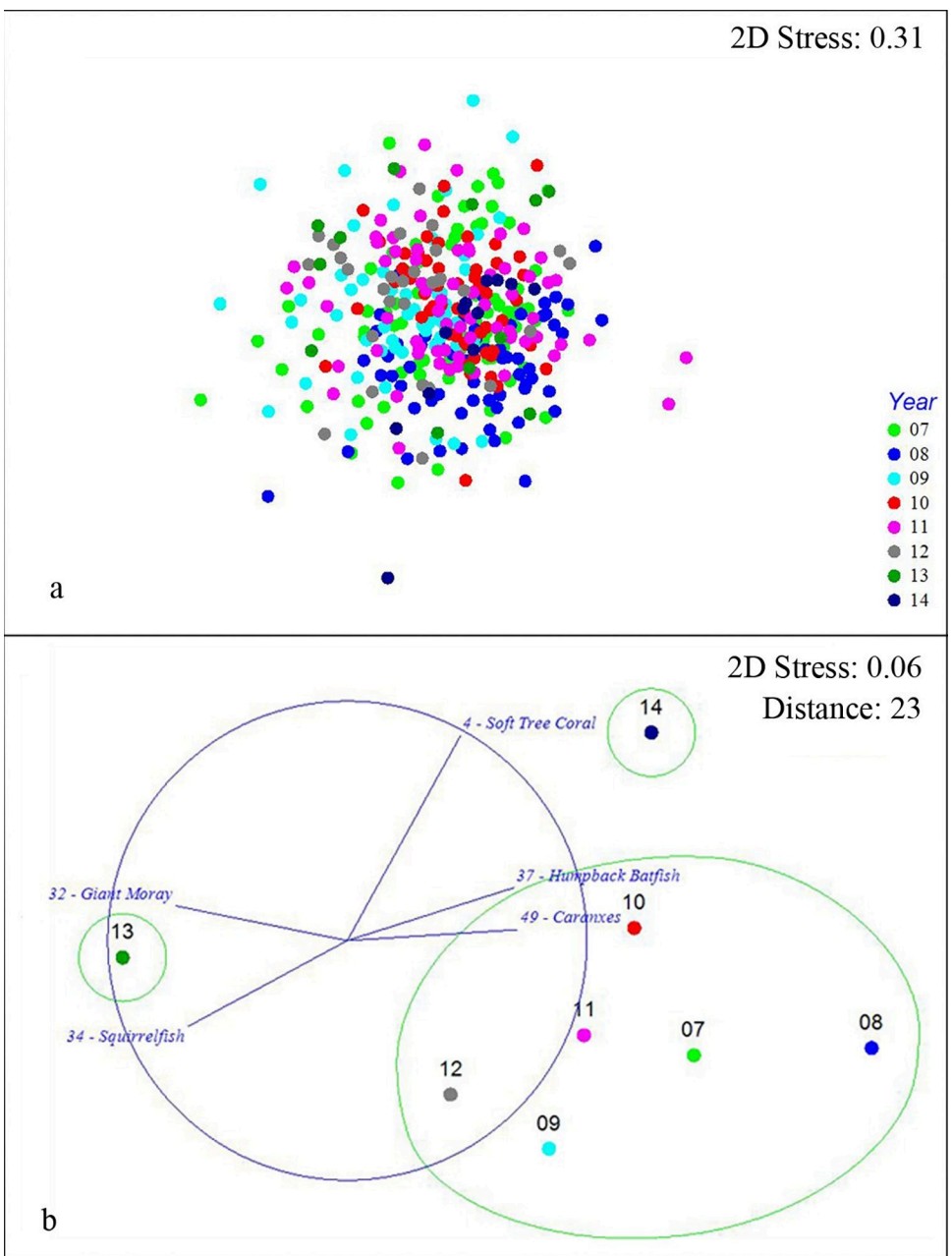

**Fig 2. a) MDS plot of the SS Thistlegorm taxa presence/absence community structure over 8 years.** Each point represents an individual questionnaire while the different shapes/colors indicate the different years. **b) Centroidal MDS of SS Thistlegorm taxa presence/absence community structure variation over 8 years with overlayed vectors of aggregated best taxa and the indicated trajectories from 2007–2014.** Each point represents the central location of the groups "Year". The green circles surround the groups resulting from the hierarchical cluster analysis.

Table 3), and Giant Moray (*Gymnothorax javanicus*) (93 in 2010 *vs* 48–78 in 2007, 2009, and 2011; Table 3) and underrepresented in Red Sea Clownfish (44 in 2010 *vs* 49–80 in 2007,2009, and 2011; Table 3). In 2012 there was an overrepresentation of Giant Moray (121 in 2012 *vs* 48–78 in 2007, 2009 and 2011; Table 3) and Napoleon Wrasse (*Cheilinus undulatus*) (55 in 2012 *vs* 26–59 in 2007,2009, and 2011; Table 3) with an underrepresentation of Soft Tree Coral (31 in 2012 *vs* 41–77 in 2007, 2009 and 2011; Table 3) and Red Sea Clownfish (35 in 2012 *vs*

**Table 3. Yearly relative abundance values of BVSTEP best species subset from centroid aggregated taxa abundance data.**

| Year | Soft Tree Coral | Giant Moray | Red Sea Clownfish | Napoleon Wrasse | Caranxes |
|------|----------------|-------------|-------------------|-----------------|----------|
| 2007 | 77 | 48 | 64 | 26 | 117 |
| 2008 | 69 | 26 | 98 | 27 | 94 |
| 2009 | 41 | 78 | 80 | 59 | 125 |
| 2010 | 98 | 93 | 44 | 37 | 154 |
| 2011 | 61 | 63 | 49 | 47 | 114 |
| 2012 | 31 | 121 | 35 | 55 | 93 |
| 2013 | 50 | 67 | 17 | 50 | 8 |
| 2014 | 154 | 62 | 31 | 0 | 115 |

49–80 in 2007, 2009 and 2011; Table 3). In 2013 there was an underrepresentation of Caranxes (Carangidae) (8 in 2013 *vs* 114–125 in 2007, 2009, and 2011; Table 3) and Red Sea Clownfish (17 in 2013 *vs* 49–80 in 2007, 2009, and 2011; Table 3). 2014 was overrepresented by Soft Tree Coral (154 in 2014 *vs* 41–77 in 2007, 2009 and 2011; Table 3) and underrepresented by Red Sea Clownfish (31 in 2014 *vs* 49–80 in 2007, 2009 and 2011; Table 3) and Napoleon Wrasse (0 in 2014 *vs* 26–59 in 2007, 2009, and 2011; Table 3).

The taxa p/a centroidal data (Fig 2B) displayed a general cluster in community structure, like the abundance data, but this general cluster was larger and represented by the years 2007 through 2012. With respect to this general cluster the year 2013 exhibited an underrepresentation in Humpback batfish (*Platax bativianus*) (33% in 2013 vs 56–86% in 2007–2012; Table 4) and Caranxes (8% in 2013 vs 52–75% in 2007–2012; Table 4). The following year showed a community similar to the general cluster but with an overrepresentation of Soft Tree Coral (92% in 2014 vs 28–54% in 2007–2012; Table 4) and an underrepresentation of Squirrel Fish (*Sargocentron spp.*) (8% in 2014 *vs* 16–52% in 2007–2012; Table 4).

## Discussion

We have described here the first investigation of temporal community trends of an historical relic, and world class dive site within the Red Sea, The SS Thistlegorm. Notwithstanding these merits, The SS Thistlegorm can also be regarded as an artificial reef community, though "accidental" in nature. Since the 1960's, artificial reefs have been used for a variety of purposes including biodiversity conservation [41], fisheries management [42], and tourist locations [43]. One study in Eilat (Red Sea) designed and submerged multiple concrete structures to test whether artificial reefs could decrease diving pressures on the surrounding natural reef areas, concluding that small scale structures, knowledge of such structures within the local diving community, and education of divers could be sufficient [44]. The provision of migratory

**Table 4. Yearly sighting frequencies of BVSTEP best species subset from centroid aggregated taxa p/a data.**

| Year | Soft Tree Coral | Giant Moray | Squirrelfish | Humpback Batfish | Caranxes |
|------|----------------|-------------|--------------|------------------|----------|
| 2007 | 46% | 38% | 33% | 70% | 55% |
| 2008 | 53% | 21% | 16% | 61% | 53% |
| 2009 | 34% | 61% | 24% | 56% | 71% |
| 2010 | 54% | 70% | 42% | 86% | 75% |
| 2011 | 44% | 47% | 38% | 61% | 65% |
| 2012 | 28% | 66% | 52% | 62% | 52% |
| 2013 | 33% | 67% | 50% | 33% | 8% |
| 2014 | 92% | 54% | 8% | 70% | 69% |

networks for coral movement and resettlement within changing oceans [45] is another application for artificial reefs, though highly debated and rather unexplored. Organismal community recruitment and interchange is an evident and well observed occurrence when any new colonizable object, be it an island or a plastic bottle, appears near a source population. In accordance with island biogeography theory [46], the evolution of this community is much dependent on the size of the object in question, the time it is in existence, and its distance from the source. Sunken warships have been shown to serve as exceptionally good artificial substrates for coral reefs as their size and complexity offer a multitude of opportunities for microhabitats, and in cases where they extend down to 30–40 m, the cooler waters can provide respite to corals from warming oceans but still sufficient light for their symbiotic zooxanthellae [23]. Many shipwrecks in the North Sea have supported hard substrate communities throughout the Belgian Continental Shelf which has otherwise been transformed into a soft-bottom environment due to anthropogenic activity [47, 48]. Thanks to the steel structure of the SS Thistlegorm, recruitment of various tolerant sessile species typical of hard substrates was enabled in the sandy bottom environment within the Straights of Gubal and a well-established coral community can now be witnessed, but no information is available in the scientific literature.

Much of the work regarding artificial coral reefs examine differences between colonization patterns on various substratum material or recently scuttled ships (less than 3 years) and natural reefs, which are largely short-term studies [24–26, 49, 50]. The few studies that have investigated community structure change over the long term (more than 5 years) have primarily been snapshot comparisons between the virgin and mature communities [51, 52] or follow succession patterns of virgin artificial reefs [53].

The present study begins to fill this gap through an 8-year monitoring of a well-established wreck community (i.e. > 80 years since the sinking of the SS Thistlegorm in 1941). Despite the internal fluctuation among years (50% in 2013, 93% in 2007), all target taxa, apart from the Manta Ray, were sighted throughout the duration of this study. These findings indicate a well-developed, if variable in time, community structure at the Thistlegorm and are in line with the monitoring of natural reefs in the region where reef sites far from touristic facilities show higher biodiversity [54]. The Kingston, a 119-year-old shipwreck 10 km east of the Thistlegorm, mimics the surrounding natural reef communities due to its similarities in structural complexity [55]. Using a visual census technique and the recordings of all species seen, oil jetties in Eilat (northern Red Sea) were studied as artificial reef proxies and were shown to have even higher species richness in fish assemblages than neighboring natural reefs [50] due to increased structural complexity. Studies of artificial reefs and wrecks in other regions throughout the world have shown similarly promising results with respect to community development. In the Florida Keys, artificial reef fish communities mirror those of natural reefs in terms of fish numbers and species composition [56] and two thirds of genera from neighboring natural reefs can thrive on sunken shipwrecks within the Caribbean [35]. With the use of SCUBA diving video sampling, microbenthic species abundances on eight shipwrecks and one airplane of varying construction and bottom type were compared to those of sampling on nearby natural substrates in the Ligurian seabed (Mediterranean Sea) [27]. The findings of this study exhibited not only a strong correlation of wreck material and bottom type to the epibenthic community present, but also revealed that some species abundances were greater on the wrecks than those of nearby natural rocky bottom substrates [27]. Even in temperate seas, such as the North Sea along the Belgian Continental Shelf, shipwrecks are described as biodiversity hotspots of hard substrate communities in an otherwise sandy bottom environment while protecting nearby soft-bottom communities from fishing activities [47, 48]. The steel structured SS Thistlegorm lies upon a sandy bottom marine environment and while this study does not

compare its taxa sightings to natural reefs nearby, it does provide key insights into the community dynamics of a well-established artificial reef.

The number of observations collected in this study through recreational citizen science methods decreased in time (Table 1). This is ascribable to the reduction in overall tourism due to political upheaval consuming Egypt during the revolution and the following transitional period of the "Arab Spring" [57, 58]. As tourists seek to vacation in more stable and safe countries, participation in citizen science is inherently affected. This temporal non-homogeneity is an evident and acknowledged limitation of citizen science, and thus it should not outweigh the benefits (e.g., increasing participant education and awareness, reduction of economic and time costs associated with environmental monitoring, and acceptable level of accuracy in data collection) of such a monitoring approach [19, 20, 54, 59].

Diving parameters were significantly correlated with the taxa data in some of the years. PY was the most frequently related dive parameter, appearing in all years with a significant relationship to diving parameters for both the taxa abundance and taxa p/a data sets (Table 2). The yearly average PY spans about half a year, ranging from early April in 2014 to late September in 2009, with little internal variation (S4 Table). Some of the variation in community structure over time in this study could thus be due to the differing seasonal environmental conditions in which questionnaires were collected. Total macroalgal biomass and community structure on coral reefs are strongly linked to seasonality and temperature variation in the Red Sea [60], where variation in benthic communities also affect the distribution of fish species [61]. Species richness, diversity, and evenness of reef fish assemblages vary significantly between summer and winter in the Red Sea [62], and these seasonal variations may be mirrored on wrecks and artificial reefs close to natural coral reefs.

Temperature, the next most linked dive parameter (Table 2) is generally influenced by both seasonality and depth. In this study, T is more likely a function of the seasonal variation because the range in average depth (AD) and Maximum Depth (MD) throughout the years is quite small (19–22 m and 25–28 m respectively) (S4 Table). In fact, coral species that have narrower depth ranges tend to occupy shallower depths, while species that can colonize deeper in the water column have wider depth ranges [63]. This may suggest that the small range of average depth in surveys at the Thistlegorm, even if statistically significant, is trivial in terms of its effect on the community structure variation given the greater depth at which the surveys were conducted. This is especially true when considering non-sessile benthic and/or free-swimming taxa. The yearly average dive duration ranged from 41 to 55 minutes (S4 Table) and presented a significant correlation in 2007 and 2008 for both data sets and in 2012 for the taxa p/a data (Table 2). Dive duration certainly plays a role in the number and amount of target taxa sighted. In validation trials of the STE project [20], volunteer scores of mean accuracies, mean consistency, and mean percent identified are all positively correlated with dive time.

Following the temporal sequence of centroids of taxa abundance (Fig 1B), no clear trend emerged. Years 2007, 2009, and 2011 cluster around what could be interpreted as a general (even if variable) community composition. In the p/a centroidal data (Fig 2B) the years 2007 through 2012 displayed a relatively larger general cluster in community structure when compared with the abundance data.

The taxa abundance data revealed more variance among years than taxa p/a data, and even if the general trend is quite similar, two of the five best species best explaining community structure variations among years are different in the analysis of the two data sets (the Red Sea Clownfish and Napoleon Wrasse for p/a data *vs*. the Squirrel Fish and Humpback Batfish for abundance data). This partial inconsistency may depend on the different reliabilities of volunteer collected data between presence/absence and abundance data sets. In fact, the correctness of abundance parameter ratings ranges from 41 to 82 percent in validation trials of the entire

STE project (and from 64 to 82 percent in the present analysis) while groups of volunteers show better performances in correctly identifying the presence of taxa [19, 20]. In 7 validation trials from the SS Thistlegorm, the parameters of data quality measurements were in line with previous analyses on coral reef environment types within the STE project [20]. The SS Thistlegorm average mean accuracy scored 57.1 percent within a range of 46 to 70 percent, coinciding with the results from [20] where 94 percent of trials scored a mean accuracy between 40 and 70 percent. From the similarity index (SI) parameter all 7 of the dives analyzed here resulted in high levels of precision (SI, 95% CI lower bound > 75% ≤ 100%), whereas only 0.4 percent of trials in [20] obtained these similarity index scores. Notwithstanding the small number of validation trials ran on the SS Thistlegorm data set with respect to the entire STE project [20], the results highly resemble those of the more rigorous trials analyzed in [20] and can therefore provide enhanced reliability to the data that has been used for evaluating the community structure variance over time at this dive site.

In conclusion, there was no clear sequential shift of the SS Thistlegorm community structure over the eight years of monitoring, but some fluctuation around a general cluster characterized by a well-developed community, mainly driven by yearly relative changes in the frequency of a few species. The temporal analysis may have been slightly biased by the different average PY (i.e., seasonality) of collected data, but this bias is likely to be minimal, as the years in which PY was significantly related to taxa data are mainly included in the general cluster and do not display unusual patterns. The community structure at the SS Thistlegorm showed relative stability over time, making this artificial reef a possible and promising refugia for Red Sea communities. Further investigation on the influence of artificial reefs as an auxiliary tool for human-influenced decline could entail the comparison of multiple wreck sites within the northern Red Sea to nearby natural coral reef dive sites through species abundance analyses between and among groupings of sites, possibly by latitudinal and/or temperature gradients.

## Supporting information

**S1 Table. Yearly sample statistics of the relate test and significance of the relation between taxa data sets (abundance and presence/absence) and diving parameters.** [*] Indicate significant differences ($P < 0.05$).
(DOCX)

**S2 Table. DistLM sequential test table of results for each year with a significant relate test between taxa abundance data and diving parameters data.** [*] Indicates significant relationship ($P < 0.05$).
(DOCX)

**S3 Table. DistLM sequential test table of results for each year with a significant relate test between taxa presence/absence data and diving parameters data.** [*] Indicates significant relationship ($P < 0.05$).
(DOCX)

**S4 Table. Yearly averages of all diving parameters with confidence intervals.** PY as date of dive (expressed as percentage of year), MD as maximum depth, AD as average depth, T as temperature, RTD as dive time, and FD as hour (expressed as percentage of day).
(DOCX)

## Acknowledgments

Special thanks go to all the divers who made this study possible.

## Author Contributions

**Conceptualization:** Erik Caroselli.

**Data curation:** Chloe Lee.

**Formal analysis:** Chloe Lee.

**Funding acquisition:** Stefano Goffredo.

**Investigation:** Chloe Lee.

**Methodology:** Chloe Lee, Erik Caroselli, Marta Meschini.

**Project administration:** Stefano Goffredo.

**Software:** Erik Caroselli.

**Supervision:** Erik Caroselli.

**Validation:** Mariana Machado Toffolo.

**Visualization:** Chloe Lee.

**Writing – original draft:** Chloe Lee.

**Writing – review & editing:** Chloe Lee, Erik Caroselli, Mariana Machado Toffolo, Arianna Mancuso, Chiara Marchini, Marta Meschini, Stefano Goffredo.

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
