## [Decision Letter · Decision Letter 0]

15 Nov 2022

PONE-D-22-26446Eight years of community structure monitoring through recreational citizen science at the “SS Thistlegorm” wreck (Red Sea)PLOS ONE

Dear Dr. Caroselli,

Thank you for submitting your manuscript to PLOS ONE. After careful consideration, we feel that it has merit but does not fully meet PLOS ONE’s publication criteria as it currently stands. Therefore, we invite you to submit a revised version of the manuscript that addresses the points raised during the review process.

 This ms has been read by two independent reviewers: both expressed appreciation but suggested change in editing and data presentation. I think the authors should not have any difficulty in addressing reviewers' requests. I also recommend the authors to revise the English carefully, possibly with the help of a mother tongue. 

We look forward to receiving your revised manuscript.

Kind regards,

Carlo Nike Bianchi

Academic Editor

PLOS ONE

Journal Requirements:

a) Did participants provide their written or verbal informed consent to participate in this study?

"STE project was funded by Project AWARE Foundation, ASTOI Association, Ministry of Tourism of the Arab Republic of Egypt, Settemari S.p.A Tour Operator, Scuba Nitrox Safety International, Viaggio nel Blu Diving Center"

"NO authors have competing interests"

7. We note that Figure 1 in your submission contain map image which may be copyrighted. All PLOS content is published under the Creative Commons Attribution License (CC BY 4.0), which means that the manuscript, images, and Supporting Information files will be freely available online, and any third party is permitted to access, download, copy, distribute, and use these materials in any way, even commercially, with proper attribution. For these reasons, we cannot publish previously copyrighted maps or satellite images created using proprietary data, such as Google software (Google Maps, Street View, and Earth). For more information, see our copyright guidelines: http://journals.plos.org/plosone/s/licenses-and-copyright.

Reviewers' comments:

Reviewer's Responses to Questions

**Comments to the Author**

1. Is the manuscript technically sound, and do the data support the conclusions?

Reviewer #1: Yes

Reviewer #2: Yes

2. Has the statistical analysis been performed appropriately and rigorously? 

Reviewer #1: I Don't Know

Reviewer #2: Yes

3. Have the authors made all data underlying the findings in their manuscript fully available?

Reviewer #1: No

Reviewer #2: Yes

4. Is the manuscript presented in an intelligible fashion and written in standard English?

Reviewer #1: No

Reviewer #2: Yes

5. Review Comments to the Author

Reviewer #1: I do not know if the english is correct, I am not an english mother tongue. I cannot download the annexes cited in the ms, hence some implementations (as new figures in the results) could be in the annexes. The same is true regarding the answer to: '..authors to make all data underlying the findings described in their manuscript fully available without restriction..? I cannot say if all data are available.

Reviewer #2: This manuscript valuably contributes to the study and the monitoring of coral communities colonising the artificial substrate of the SS Thistlegorm wreck, in the Red Sea. Through citizens science, the involvement of enthusiasts volunteer recreational divers allowed substantial data collection for a relatively long time. The results give hope that projects like this will last over time.

I have no major concerns to address to the authors. Please see minor comments and corrections as comments in the .pdf file of the manuscript.

I suggest the publication of the manuscript after minor revisions.

6. PLOS authors have the option to publish the peer review history of their article (what does this mean?). If published, this will include your full peer review and any attached files.

Reviewer #1: **Yes: **Andrea Peirano

Reviewer #2: No

---

## [Author Response · Author response to Decision Letter 0]

18 Jan 2023

Response to Reviewers

We thank the reviewers for their constructive comments which allowed us to improve the scientific quality and clarity of the manuscript. We agreed with all suggestions and edited the manuscript accordingly. A point-by-point reply to each comment is provided below.

Reviewer #1 

COMMENT - I do not know if the English is correct, I am not an English mother tongue.

REPLY – We have had the manuscript checked for sentence structure and grammar by a native English speaker.

COMMENT - I cannot download the annexes cited in the ms, hence some implementations (as new figures in the results) could be in the annexes. 

REPLY – We hope the supporting information is available to download in the current version. 

COMMENT - The same is true regarding the answer to: '..authors to make all data underlying the findings described in their manuscript fully available without restriction..? I cannot say if all data are available.

REPLY – All raw data will be made available upon publication of the paper.

COMMENT – “dive” deletion (pg 1, Keywords).

REPLY – Agreed. the word dive was deleted from the keywords in the manuscript. 

COMMENT – “www.steproject.org” deletion (pg 2, Abstract).

REPLY – Agreed. This was deleted from the manuscript.

COMMENT – I suggest to write the taxa (pg 2, Abstract). 

REPLY – Agreed. The five main taxa of both datasets have been added.

COMMENT – Materials? Experiential benefits? Explain better (pg 3, pp 1, Introduction).

REPLY – Agreed. Examples have been added to the sentence for a clearer explanation. 

COMMENT – I suggest to break the sentence into “…. STE program launched in 2007 (www.steproject.org)” (pg 4, pp 1, Introduction).

REPLY – Agreed. The suggested modification of the sentence was added to the manuscript.

COMMENT – I suggest to change in some sentences the term volunteers into recreational divers (pp 4, pp 1, Introduction).

REPLY – Agreed. Some of the terms “volunteers” were changed to divers or participants.

COMMENT – Delete “of their choosing” (pg 4, pp 1, Introduction).

REPLY – Agreed. This phrase was deleted for a more concise sentence. 

COMMENT – What does no behavioral change of the divers mean? (pg 4, pp 1, Introduction).

REPLY – Agreed. The following sentence was added to explain: This approach to citizen science allows participants to carry out their normal activities (volunteer behavior is unchanged throughout the survey), and the collection of casually observed data. 

COMMENT – What does it mean, please explain? Regarding “based on correlations to reference researchers” (pg 4, pp 1, Introduction).

REPLY – Agreed. It has been explained that the validation trials were ran by comparing volunteer collected data to those of data collected by “control divers” (marine biologists).

COMMENT – What is Consistency, please explain (pg 4, pp 1, Introduction).

REPLY – Agreed. The extrapolation “Consistency, or the similarity of data collected by individual volunteers during the same dive” was added to the manuscript.

COMMENT – What is Percent Identified, please explain (pg 4, pp 1, Introduction).

REPLY – Agreed. The sentence “The percentage recorded of the total number of taxa present (acquired from control diver data), or the Percent Identified, was the highest-ranking parameter” was added to the manuscript. 

COMMENT – I suggest “however” instead of “overall” (pg 4, pp 1, Introduction).

REPLY – Agreed. We have modified the text according to the suggestion. 

COMMENT – In this part, the wreck study should be supported by an introduction of the importance of wreck studies for communities and/or recovery species. I suggest the readings of paper Peirano (2013) and references therein (pg 4, pp 1, Introduction).

REPLY – Agreed. The importance of wreck studies for communities was discussed within the Discussion of the manuscript and has now been moved to the Introduction where Peirano (2013) was also included as a source along with additional sources from that publication. 

COMMENT – Please rearrange the sentences describing the ship. What is important is the geographical position, the depth, the height, and the length (pg 4, pp 2, Introduction). 

REPLY - Agreed. The sentence was made shorter and more concise with the suggested dimensions added. 

COMMENT – The ship was heavily damaged also by plundering of souvenirs by recreational divers (pg 5, pp 1, Introduction).

REPLY – Agreed. An additional sentence was added to the manuscript to describe the destruction of the SS Thistlegorm by looting. 

COMMENT – Far? How many miles far? Today you can visit the wreck with boats that can reach it within a few hours. (pg 5, pp 2, Introduction).

REPLY – Agreed. The distance from the main resorts in the Sharm el Sheikh area has been added to the manuscript.

COMMENT – I think that all the part evidenced in green should be used in the conclusion rather than the introduction (pg 5, pp 2, Introduction).

REPLY – Agreed. This section has been moved to the discussion. 

COMMET – deletion of “one of the most top-ranking dive sites in the world (pg 6, pp 2, Introduction).

REPLY – Agreed. This was removed from the manuscript.

COMMENT – delete “data were collected within” and “whose efforts” (pg 6, pp 3, Materials and Methods – Data collection and isolation).

REPLY – Agreed. These were deleted to form a more concise sentence. 

COMMENT – Please explain, regarding “the behavior of underwater tourists is unaltered” (pg 6, pp 3, Materials and Methods – data collection and isolation).

REPLY – Agreed. Further details of this concept were added when it was introduced within the Introduction of the manuscript.

COMMENT – Change “participant volunteers” to “recreational SCUBA divers” (pg 6, pp 3, Materials and Methods – data collection and isolation).

REPLY – Agreed. We have modified the text according to the suggestion.

COMMENT – Introduce here the codes used in the results (T, RTD, PY etc.) (pg 6, pp3, Materials and Methods – Data collection and Isolation).

REPLY – Agreed. The codes were have been introduced within the materials and methods section. 

COMMENT – Sightings of what? Fishes, coral? Marked on tablets? Please give some explanations, this is the core of the methods. (pg 6, pp 3, Materials and Methods – Data collection and Isolation).

REPLY – Agreed. The sentence “The 72 faunal taxa were chosen because they are representative of the main ecosystem trophic levels within the Red Sea, they are common/abundant, and they are easily identifiable by recreational divers” Was added to the manuscript. It was also explained that the sightings were recorded on the questionnaires immediately following the dive without the use of tablets or recording material during the dive. 

COMMENT – Deletion of “for detailed methods, see..” (pg 6, pp 3, Materials and Methods – Data collection and isolation).

Reply – Agreed. This was deleted from the manuscript and the methods were explained in more detail within. 

COMMENT – I think that the shipwreck could be introduced at the start of the materials and methods section as Study Site. It should describe the wreck, its position on the bottom, type of bottom, presence of reefs in the surroundings, currents, type of construction (particularly that it is full of cars, motorbike, explosives etc that could be harmful to the marine environment. (pg 6, pp 3, Materials and Methods – data collection and isolation).

REPLY – Agreed. A new subsection titled “The Study Site” under materials and methods was added to the manuscript and described the location, cargo, currents, type of construction, and substrate. 

COMMENT – Deletion of “were isolated from the entire STE data set”. (pg 6, pp 3, Materials and Methods – data collection and isolation).

REPLY – Agreed. This was deleted from the manuscript.

COMMENT – This section along with Table 1, should be moved to the Results (pg 6, pp 3, Materials and Methods – data collection and isolation). 

REPLY – Agreed. The section regarding number of questionnaires used or removed from the study was moved to the Results section. 

COMMENT – It is strange this ratio (pg 7, pp 2, Materials and Methods – Preliminary analysis and treatment).

REPLY – Agreed. The equation has now been inverted to its correct form in the manuscript. 

COMMENT – I do not understand how it is calculated the average sighting abundance (pg 7, pp 2, Materials and Methods – Preliminary treatment and analysis).

REPLY – Agreed. The description “For each of the taxa sighted, the divers also recorded an estimated Sighting Abundance according to 3 classes: 1, 2, and 3 (rare, frequent, and very frequent respectively). The classes were weighted to each taxon’s individual expected occurrence” was added to the Data collection and isolation section of the Materials and Methods. 

COMMENT – Deletion of BVSTEP (pg 7, pp 3, Materials and Methods – preliminary analysis and treatment).

REPLY – Agreed. This was deleted from the citation.

COMMENT – Deletion of Relate (pg 7, pp 4, Materials and Methods – preliminary analysis and treatment).

REPLY – Agreed. This was deleted from the citation. 

COMMENT – Deletion of DistLM (pg 7, pp 5, Materials and Methods – preliminary analysis and treatment).

REPLY – Agreed. This was deleted from the citation.

COMMENT – This sentence should introduce (at the start) of the description of the analysis (pg 8, pp 3, Materials and Methods – preliminary analysis and treatment).

REPLY – Agreed. The sentence was moved to the beginning of the section. 

COMMENT – This paragraph should be moved to the beginning of preliminary analysis and treatment (pg 8, pp 4, Materials and Methods – preliminary analysis and treatment)

REPLY – Agreed. The paragraph was moved to the beginning of the preliminary analysis and treatment section of the manuscript. 

COMMENT – What does it mean (regarding “best subset of taxa”) (pg 8, pp 5, Materials and Methods – temporal analysis). 

REPLY – Agreed. The sentence was modified to “…only the BVSTEP best subsets of taxa representing the community structure from any individual year were obtained…”.

COMMENT – Why “again”? (pg 8, pp 5, Materials and Methods – temporal analysis).

REPLY – Agreed. The word was deleted from the manuscript. 

COMMENT – Deletion of “Bray-Curtis similarity” from citation (pg 8, pp 5, Materials and Methods – temporal analysis).

REPLY – Agreed. This was deleted form the manuscript. 

COMMENT – Deletion of “CLUSTER” from citation (pg 8, pp 5, Materials and Methods – temporal analysis).

REPLY – Agreed. This was deleted form the manuscript. 

COMMENT – Deletion of “BVSTEP” from citation (pg 8, pp 5, Materials and Methods – temporal analysis).

REPLY – Agreed. This was deleted form the manuscript. 

COMMENT – Add significance level after “correlated” (pg 9, pp2, Results – preliminary analysis and treatment).

REPLY – Agreed. Significance level was added just after “correlated” and deleted from the end of the sentence.

COMMENT – Add significance level after “significantly related” (pg 9, pp2, Results – preliminary analysis and treatment).

REPLY – Agreed. Significance level was added just after “significantly related” and deleted from the end of the sentence.

COMMENT – Change p to P (pg 10, table 2 caption, Results – preliminary analysis and treatment).

REPLY – Agreed. “p” was changed to “P” in the table caption.

COMMENT – Of what? Regarding “abundance and presence/absence” data (pg 10, table 2 caption, Results – preliminary analysis and treatment).

REPLY – Agreed. “Taxa” abundance and “taxa” presence/absence data were added to the Table 2 caption. 

COMMENT – Percentage of what? Previously PY was described as date (pg 10, Table 2 caption, Results – preliminary analysis and treatment).

REPLY – Agreed. The sentence was changed to “… PY as date (expressed as the percentage of year) …”.

COMMENT – Percentage of what? Previously FD was described as hour (pg 10, Table 2 caption, Results – preliminary analysis and treatment).

REPLY – Agreed. The sentence was changed to “… FD as hour (expressed as the percentage of day) …”. 

COMMENT – Presence/absence? Regarding “p/a” (pg 10, pp 1, Results – Preliminary analysis and treatment).

REPLY – Agreed. “p/a” was redefined in this section.

COMMENT – Change “Relate” to “RELATE” (pg 10, pp1, Results – Preliminary analysis).

REPLY – Agreed. The suggested modification was changed within the manuscript. 

COMMENT – You mean “in” 2007, not “from” 2007 (pg 10, pp 1, Results – preliminary analysis and treatment).

REPLY – Agreed. “In” was changed to “from” in the manuscript. 

COMMENT – you present data on taxa abundance and community structure. however, as written in one note in the material and methods part, you did not write about what have you considered as targets ( fish ?, corals?) and what you means as community structure.For example a coral community is formed by benthic corals and fishes.Caranxes are pelagic fishes.You have done a great work, please spend some time to clear all these points.(pg 10, pp 4, Results – Temporal analysis). 

REPLY – Agreed. An explanation of how the target taxa were chosen has been added to the manuscript (i.e. The 72 faunal taxa were chosen because they are representative of the main ecosystem trophic levels within the Red Sea, they are common/abundant, and they are easily identifiable by recreational divers).

COMMENT – One species? Regarding the BVSTEP best species of centroidal data (pg 11, pp 2, Results – temporal analysis).

REPLY – Agreed. The scientific names of the species/genre/families were added to the manuscript. 

COMMENT – This part could be used in the Introduction (pg 12, pp 1, Discussion).

REPLY – Agreed. This part, involving the discussion of the importance of wrecks to marine communities was moved to the Introduction upon earlier request. 

COMMENT – I do not know if in your data we can talk about diversity, if so, you can show this in the results (pg 13, pp 2, Discussion).

REPLY – Agreed. Diversity is no longer discussed in reference to our data but only in reference to external sources. 

COMMENT – I think you should consider the different approach of your study with touristic divers and the species list proposed for the cited papers. I should be cautious in your species richness (we don’t have the data in you MS) with cited data (pg 13, pp 2, Discussion). 

REPLY – Agreed. The methodology of the paper cited regarding this comment was described and added to the manuscript.

COMMENT – Also in Mediterranean there are studies on this subject (pg 13, pp 2, Discussion)

REPLY – Agreed. Additional sources have been added to describe studies that have taken place within the Mediterranean.

COMMENT – Incorrect spelling of “correlated” (pg 14, pp 2, Discussion).

REPLY – Agreed. The misspelling was corrected.

COMMENT – All of the following parameters are interesting, however, to talk about them they should be proposed in results at least in figures/tables (I cannot see the appendix). I think you should propose figures showing the fluctuation of recorded parameters (pg 14, pp 3, Discussion).

REPLY – Agreed. These tables are already included in the supporting Information file of the manuscript which was unavailable for you to download at the time. We hope that it is made available in the current version. 

COMMENT – I do not know the species you are talking about however I do not think that some shallow water species cannot be found at 30 m depth. Maybe the distance from the coast, the type of substratum, or currents may influence the colonization. But you agree with me in the following sentences? Or not? (pg 14, pp 3, Discussion).

REPLY – Agreed. We believe the sentences following the statement agree with your supposition. If corals can occupy deeper in the water column, they usually have a greater depth range of colonization. The sentence was rearranged to “In fact, coral species that have narrower depth ranges tend occupy shallower waters…” to clarify the statement. 

COMMENT – I think that the following sentences evidenced in green should be moved to the results (pg 15-16, pp 2 and 3, pp 2, Discussion).

REPLY – The suggested sections were moved to the results.

Reviewer #2

COMMENT – Maybe add citizen science to the keywords (pg 1, Keywords).

REPLY – Agreed. The term “citizen science” was added to Keywords. 

COMMENT – I think that it is more appropriate to use the term “resistance” because, as you said, there have been few bleaching events thus corals have resisted to temperature anomalies (pg 3, pp 2, Introduction).

REPLY – Agree. The term “resilience” was replaced with “resistance”. 

COMMENT – We do not “use” citizen science, we practice it. It is a way of doing citizen science and therefore cannot be considered as a tool (pg 3, pp 3, Introduction).

REPLY – Agreed. The term “use” was changed to “practice” in the manuscript. 

COMMENT – The term stakeholder should be changed to the plural form (pg 4, pp 1, Introduction).

REPLY – Agreed. The word was changed to plural. 

COMMENT – You are talking about people, not objects. Better to use a more appropriate synonym regarding the term “utilized” (pg 4, pp 1, Introduction).

REPLY – Agreed. The term “utilized” was replaced with “involving”.

COMMENT – Add “.” (pg 4, pp 2, Introduction).

REPLY – Agreed. “.” Was added to the manuscript. 

COMMENT – Add “.” (pg 6, pp 1, Introduction).

REPLY – Agreed. “.” Was added to the manuscript.

COMMENT – Please cite the reference number in square brackets (pg 14, pp 3, Discussion).

REPLY – Agreed. The reference number in brackets was added to the manuscript.

COMMENT – Please cite the reference number in square brackets (pg 14, pp 3, Discussion).

REPLY – Agreed. the reference number in brackets was added to the manuscript. 

COMMENT – This section sounds like results and all those numbers make it hard to read. Please move them to the results and limit here the discussion of those results (pg 15, pp 2, Discussion)

REPLY – Agreed. Most of the section selected was moved to the results section. 

COMMENT – Remove “best” (pg 16, pp 2, Discussion).

REPLY – Agreed. The sentence was restructured to define the BVSTEP Best test.

---

## [Editor Report · Decision Letter 1]

10 Feb 2023

Eight years of community structure monitoring through recreational citizen science at the “SS Thistlegorm” wreck (Red Sea)

PONE-D-22-26446R1

Dear Dr. Caroselli,

We’re pleased to inform you that your manuscript has been judged scientifically suitable for publication and will be formally accepted for publication once it meets all outstanding technical requirements.

Kind regards,

Carlo Nike Bianchi

Academic Editor

PLOS ONE
---

## [Editor Report · Acceptance letter]

17 Feb 2023

PONE-D-22-26446R1 

Eight years of community structure monitoring through recreational citizen science at the “SS Thistlegorm” wreck (Red Sea) 

Dear Dr. Caroselli:

I'm pleased to inform you that your manuscript has been deemed suitable for publication in PLOS ONE. Congratulations! Your manuscript is now with our production department. 

Kind regards, 

on behalf of

Dr. Carlo Nike Bianchi 

Academic Editor

PLOS ONE